# *Arabidopsis* shoot stem cells display dynamic transcription and DNA methylation patterns

Ruben Gutzat[1],[*] iD, Klaus Rembart[1], Thomas Nussbaumer[2],[†],[‡], Falko Hofmann[1] iD, Rahul Pisupati[1] iD, Gabriele Bradamante[1], Nina Daubel[1],[§], Angelika Gaidora[1],[¶], Nicole Lettner[1], Mattia Donà[1], Magnus Nordborg[1], Michael Nodine[1] & Ortrun Mittelsten Scheid[1],[**] iD

## Abstract

In plants, aerial organs originate continuously from stem cells in the center of the shoot apical meristem. Descendants of stem cells in the subepidermal layer are progenitors of germ cells, giving rise to male and female gametes. In these cells, mutations, including insertions of transposable elements or viruses, must be avoided to preserve genome integrity across generations. To investigate the molecular characteristics of stem cells in *Arabidopsis*, we isolated their nuclei and analyzed stage-specific gene expression and DNA methylation in plants of different ages. Stem cell expression signatures are largely defined by developmental stage but include a core set of stem cell-specific genes, among which are genes implicated in epigenetic silencing. Transiently increased expression of transposable elements in meristems prior to flower induction correlates with increasing CHG methylation during development and decreased CHH methylation, before stem cells enter the reproductive lineage. These results suggest that epigenetic reprogramming may occur at an early stage in this lineage and could contribute to genome protection in stem cells during germline development.

**Keywords** DNA methylation; fluorescence-activated nuclei sorting; plant development; stem cells; transcriptome
**Subject Categories** Chromatin, Transcription & Genomics; Development; Plant Biology
**The EMBO Journal (2020) 39: e103667**

## Introduction

Active transposable elements (TEs) are a severe threat to genome integrity in all organisms, especially in germline cells that form gametes and subsequent generations. Strategies for suppressing TEs differ in animals and plants, based on their respective life cycles. In animals, the body plan is fixed during embryonal development, including an early specification of germ cell precursors. During an animal's lifetime, specialized stem cell populations support tissue and organ regeneration. In contrast, many plant cells remain totipotent, and plants develop organs *de novo* during their life, with constant feedback from, and adjustment to the environment. As many TEs are mobilized by external triggers, the risk of insertions that affect subsequent generations is generally much higher in plants.

All mobile elements require an RNA intermediate for their propagation. Host defenses exploit this dependency, by transcriptionally inactivating the genes necessary for transposition, via epigenetic modifications such as DNA methylation and heterochromatin formation. RNA-directed DNA methylation (RdDM) is a central element of TE control in plants (reviewed, e.g., in Cui & Cao, 2014; Wendte & Pikaard, 2017). Many plant proteins involved are encoded by large gene family members and have diversified and specialized in function (reviewed in Xie *et al*, 2004; Lee & Carroll, 2018). Evolutionary theory predicts selective pressure on TEs to exclusively proliferate in stem cells that can develop into gametes and gametes themselves (Haig, 2016). However, TE activation and DNA methylation at their genes are usually analyzed in somatic tissue, in which new TE copies are rarely inherited by progeny (e.g., Hirochika *et al*, 2000; Lippman *et al*, 2003; Tsukahara *et al*, 2009). Nevertheless, active TEs can multiply quickly between generations if epigenetic control is transiently repressed (e.g., Ito *et al*, 2010; Marí-Ordóñez *et al*, 2013), consistent with evidence suggesting a specific role for epigenetic regulation of repetitive elements in germline formation in the shoot apical meristem (SAM) (Baubec *et al*, 2014).

The center of the SAM contains stem cells responsible for growth and formation of all non-embryonic, above-ground organs, including those for reproduction. In *Arabidopsis*, they are defined by expression of *CLAVATA3 (CLV3)* (Laux, 2003). Genetic changes

---

1  Gregor Mendel Institute (GMI), Austrian Academy of Sciences, Vienna BioCenter (VBC), Vienna, Austria
2  Division of Computational Systems Biology, Department of Microbiology and Ecosystem Science, University of Vienna, Vienna, Austria
   *Corresponding author. Tel: +43 1 790449831; E-mail: ruben.gutzat@gmi.oeaw.ac.at
   **Corresponding author. Tel: +43 1 790449830; E-mail: ortrun.mittelstenscheid@gmi.oeaw.ac.at
   †Present address: Chair and Institute of Environmental Medicine, UNIKA-T, Technical University of Munich and Helmholtz Zentrum München, Augsburg, Germany
   ‡Present address: Institute of Network Biology (INET), Helmholtz Center Munich, Neuherberg, Germany
   §Present address: Uppsala University, Uppsala, Sweden
   ¶Present address: University of Natural Resources and Life Sciences, Vienna, Austria

transmitted to the next generation, including new TE insertions, are expected to pass through the genome of these stem cells. Whether this requirement presents a bottleneck for TE amplification, whether the stem cells have a specialized or enforced TE silencing mechanism, and how such a control could be achieved are open questions.

Despite their essential role for plant development and intergenerational continuity, a comprehensive molecular analysis of SAM stem cells throughout development is lacking, due to their small number, deeply embedded among non-stem cells, making them difficult to isolate. To overcome these limitations, we have collected pure SAM stem cell nuclei at different stages of the *Arabidopsis* life cycle and combined transcriptome profiling with genome-wide DNA methylation analysis. The results reveal a small number of genes of the epigenetic control system that are preferentially expressed in stem cells and a transient activation of specific TEs prior to flower induction. Dynamic DNA methylation at TEs indicates that epigenetic reprogramming occurs preceding gamete formation. These mechanisms could contribute to a reinforced "quality control" system for faithful transmission of genetic and epigenetic information.

# Results

### Purification of SAM stem cell nuclei

To develop a robust protocol suitable for stem cell nuclei preparation across all *Arabidopsis* developmental stages, we generated plants expressing mCherry-labeled histone H2B under control of the stem cell-specific *CLV3* promoter (Tucker *et al*, 2008). Microscopy of 14-day-old seedlings was used to demonstrate the expected fluorescence signals in nuclei of $\approx$ 20–40 stem cells (Fig 1A). We applied fluorescence-activated nuclear sorting (FANS) (Zhang *et al*, 2008) to nuclei isolated from manually enriched SAMs and collected mCherry-positive and mCherry-negative nuclei, with non-transgenic plants as controls (Figs 1A and EV1A, Appendix Table S1) (Gutzat & Mittelsten Scheid, 2020). Microscopy revealed that all nuclei sorted into the positive channel appeared intact and displayed red fluorescence, validating the purity of the fraction (Fig EV1AB). High levels of endogenous *CLV3* transcript in mCherry-positive (> 1,000-fold) versus controls (Fig 1C) confirmed enrichment of stem cell nuclei. To assess whether nuclear RNA was an adequate proxy for the whole transcriptome, we compared RNA-seq data between libraries from whole seedlings and those from sorted nuclei. The high correlation (Pearson correlation coefficient for all genes = 0.9; Fig EV2) indicated that nuclear RNA from the pure fractions of stem cell nuclei is representative of the transcriptome of whole cells, including TEs and pseudogenes.

### Stem cell developmental expression signatures

We generated and sequenced RNA expression libraries from stem and non-stem cell nuclei isolated from manually isolated embryos (heart through torpedo stage, E), and plants 7, 14, or 35 days after germination, corresponding to juvenile plants with vegetative growth (D7), adult plants switching from vegetative to reproductive growth (D14), or flowering plants (D35), respectively (Fig EV3A–C, Dataset EV1 & Table EV1). Expression data were highly

reproducible between replicates, and housekeeping genes (Czechowski *et al*, 2005) were uniformly represented across all samples and all stages. The data were then scored for differentially expressed genes (DEGs) in stem cells, dependent and independent of development. Expression of *CLV3* and *mCherry* transcripts exclusively in stem cell nuclei was confirmed at all developmental stages (Fig 2A). Transcripts for *TEL2* and *PAN*, two genes expressed in the central domain of the SAM (Anderson *et al*, 2004; Maier *et al*, 2011), and the meristem marker genes *STM* and *KNAT1* (Lincoln *et al*, 1994; Long *et al*, 1996) were present at high levels in stem cell nuclei, confirming their meristematic nature, and at lower levels in cells surrounding the stem cells, and their expression was very low in samples from 14-day-old whole seedlings (Fig 2A). Correlation analysis of all samples showed that the expression signature of stem cells was dominated by developmental stage rather than stem cell character (Fig 2B), consistent with the plastic and environment-dependent development of plants.

Comparing all stem cell samples with all non-stem cell samples yielded 86 DEGs with increased, and seven DEGs with decreased expression, both groups enriched for transcription factors (*P*-value < 1.0e−08; Fig 2C and Dataset EV1). We detected a limited but significant overlap for up-regulated genes with the meristem transcriptome of the *ap1-1;cal1-1* double-mutant (Yadav *et al*, 2009) and that of 7-day-old seedlings (Tian *et al*, 2019) (Table EV2). Comparison with transcriptome data for different types of root meristem cells (de Luis Balaguer *et al*, 2017) revealed an overlap with up-regulated genes in *WOX5*-expressing cells of the quiescent center (Table EV2), indicating that the WOX5 stem cell niche (SCN) is functionally closer to the CLV3 SAM domain than the root SCN.

Pairwise comparison between stem cell nuclei and non-stem cell nuclei from the same stage revealed that the majority of DEGs (*q* < 0.05) were up-regulated, except for the embryo samples (Fig 2C). In addition to genes involved in meristem maintenance and general regulation of gene expression, we also found significant enrichment of up-regulated genes related to shoot system and flower development, even at early time points E and D7 (Table EV3, Appendix Fig S1) indicating that formation of the transcriptional signatures in SAM stem cells precedes later developmental processes. Transcription factors are enriched among up-regulated DEGs at all four time points (hypergeometric test, $P \leq 0.01$), suggesting that mainly positive control determines the transcriptome in SAM stem cells. Taken together, the expression profile of stem cells in *Arabidopsis* varies with development and does not present a general particular molecular signature at all stages, with the exception of a few stem cell-specific genes. To identify these, we examined the overlap of DEGs from the pairwise comparison between the stem cell and the respective non-stem cell libraries across the four time points. Thirty-two genes, including *CLV3*, were more highly expressed in stem cell nuclei in at least three of the four stages, and nine of these DEGs are shared across all time points (Fig 2D, Appendix Figs S2A–C and S3, Table EV4). Significant GO terms for this set of genes include "reproductive shoot system development" and "flower development", in addition to the expected categories "meristem maintenance" and "meristem development" (Fig 2D), similar to the DEGs in individual sample pairs (described above). Here, we focus specifically on the epigenetic control of TEs in the stem cells and therefore consider only gene families for epigenetic regulators among the DEGs. We found significantly elevated

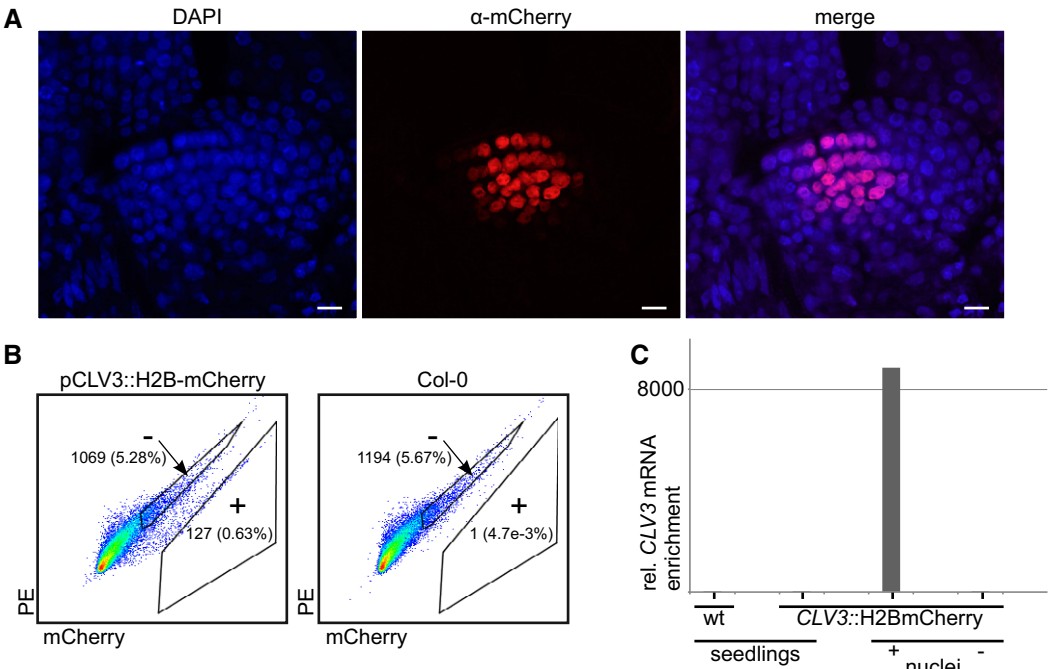

**Figure 1. Establishment of FANS for stem cells of the shoot apical meristem (SAM).**

A　Expression of H2B-mCherry under control of the *CLV3* promoter in 14-day-old seedlings. Whole-mount immunostaining using α-mCherry antibodies and laser scanning microscopy (scale bar 10 μm).

B　Example of a FANS experiment: mCherry-positive (+) and mCherry-negative (−) gates of DAPI-gated nuclei. Numbers indicate total number and percent of DAPI (for −) and mCherry (for +) events.

C　A representative example for enrichment of *CLV3* transcript in mCherry-positive nuclei determined by qRT–PCR and normalized to wt (*N* = 1).

expression of several silencing-related genes, described below. The remaining genes specifically expressed in stem cells are discussed in more detail in the Appendix Supplementary Text.

**Silencing-related genes are up-regulated in SAM stem cells**

We assembled a list of 62 genes associated with a role in epigenetic regulation, based on previous reports (Stroud *et al*, 2013; and many others). Ten of those were identified as stem cell DEGs up-regulated during one or more time point (*P*-value of enrichment 6.03e−10; Fig 3A). These included two Argonaute proteins (AGO5 and AGO9), two histone methyltransferases (SUVH4 and SUVR2), the nucleosome remodeler DDM1, the histone variant H2A.W7, a subunit of PolIV (NRPD1a), and two putative RNA-dependent RNA polymerases (RDR4 and 5) (Fig 3B) (Yu *et al*, 2003). AGO9 is a member of clade III of the Argonaute proteins in *Arabidopsis* (Vaucheret, 2008) and is involved in TE repression during gametophyte development and DNA repair (Duran-Figueroa & Vielle-Calzada, 2010; Havecker *et al*, 2010; Olmedo-Monfil *et al*, 2010; Oliver *et al*, 2014). AGO5 belongs to clade II, and its role in gene silencing pathways is less clear (Mi *et al*, 2008; Tucker *et al*, 2012; Brosseau & Moffett, 2015). RDR4 and 5 were named according to their protein sequence similarity with RDR2 and 6, central components of the RdDM pathway, but have uncharacterized function. DDM1, SUVH4, SUVR2, and NRPD1a are all expressed in SAM stem cells across the four different stages, but at varying levels, and all have roles in establishment and maintenance of DNA methylation and heterochromatin

required for repressing transcription of TEs (Pikaard & Mittelsten Scheid, 2014). H2A.W7 is a plant- and heterochromatin-specific variant of histone 2A (Lorkovic *et al*, 2017) and likely involved in TE control. Thus, SAM stem cell transcriptomes are enriched for components of epigenetic silencing, suggesting tightened control of TEs in SAM stem cells throughout development.

**Dynamic expression of TEs in SAM stem cells**

To test the idea of reinforced control of TEs, we analyzed the stem cell transcriptome data for expression of TEs. We calculated expression differences between SAM stem cells and surrounding cells at each time point for individual TEs for which the sequence reads allowed unambiguous mapping to the genome. Although TE expression was generally low, it increased in stem cells at the vegetative stage (D7 Fig 4A, Appendix Figs S4 and S5). This increase is specific for TEs, as housekeeping genes (Czechowski *et al*, 2005) gave constant values at all time points (Fig 4A). In D7 samples, 59 out of 62 differentially expressed TEs were more highly expressed in stem cells than surrounding cells (*P*-value < 0.05, not sample size adjusted) (Dataset EV2 and Appendix Fig S5). Thirty of these belong to the LTR/COPIA and LTR/GYPSY super-families and include 14 TEs of families that are known to be mobile (Dataset EV2, sheet D7-TE-genes) (Quadrana & Colot, 2016). To extend the analysis to highly repetitive TEs that cannot be mapped to unique positions, we further assigned sequencing reads to whole TE families with the TETool kit (Jin *et al*, 2015). This analysis, including 318 class I and

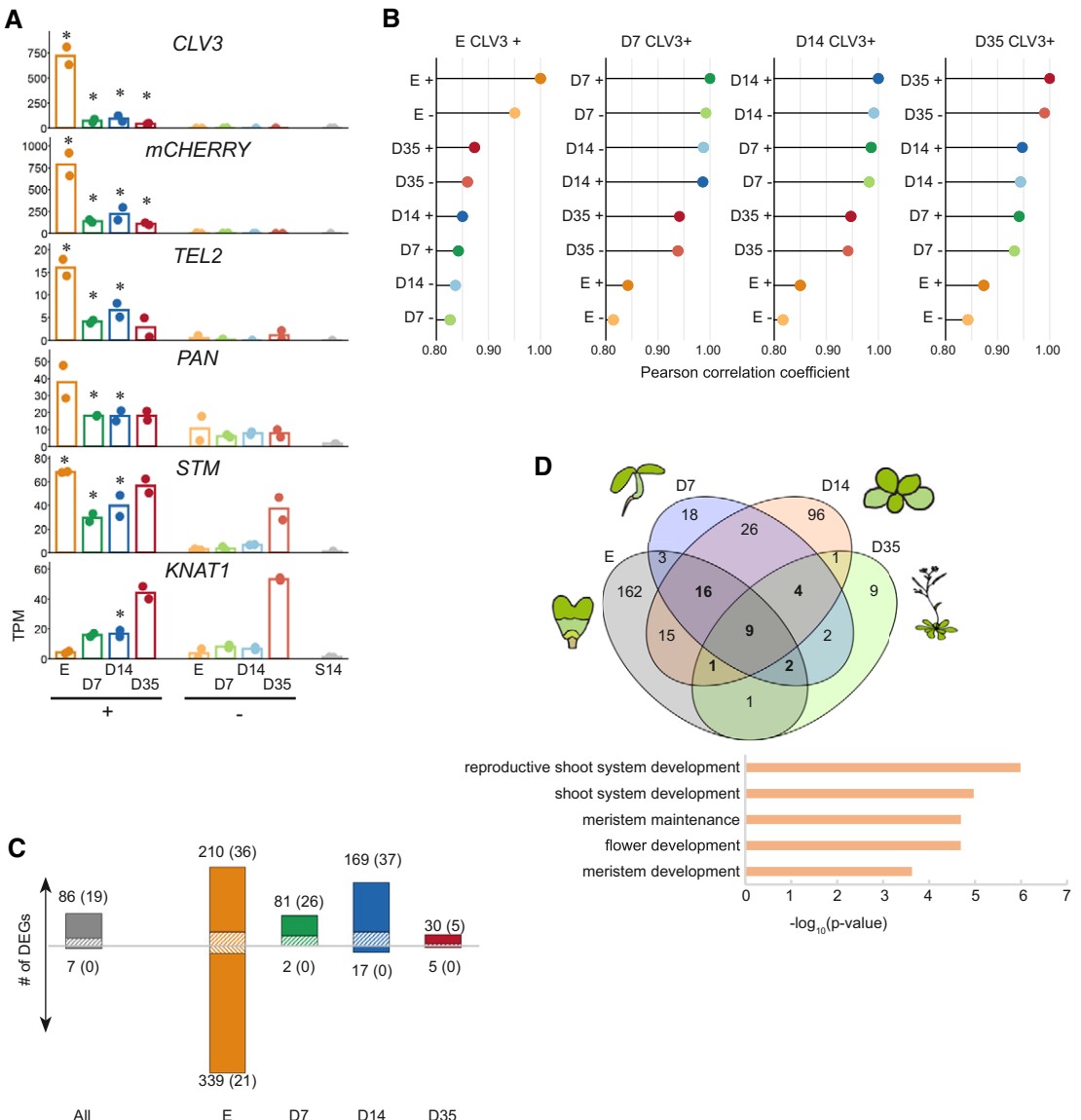

**Figure 2. Differential RNA expression in SAM stem cells during development.**

A Expression of *CLV3*, *mCherry*, *TEL2*, *PAN*, and the meristem marker genes *STM* and *KNAT1*. Asterisks indicate time points of significantly different expression (Wald test, Benjamini & Hochberg corrected, *q* < 0.05) between stem and non-stem cells for each time point (N = 2). + = stem cell nuclei; − = non-stem cell nuclei, E = nuclei from embryos, D7/14/35 = nuclei from 7/14/35-day-old plants, S14 = nuclei from 14-day-old above-ground seedlings.

B Global gene expression correlations at different developmental stages of stem (+) and non-stem cell (−) nuclei.

C Number of DEGs between stem and non-stem cells at each time point. The banded portion of the bars indicates the number of transcription factor genes (also in parenthesis).

D Overlap of genes with higher expression in stem cells (excluding *mCherry*) and significant GO terms for shared DEGs (increased expression during at least three time points). See also Appendix Fig S2 for *P*-values of overlapping gene sets. *P*-values were calculated with Fisher's exact test and Bonferroni-corrected (see also Table EV3).

class II TE families, resulted in a similar pattern as the single TEs: Expression was reduced in E and D35 stem cells compared to non-stem cells, in contrast to 54 families with increased expression in vegetative (D7) SAM stem cells (Fig 4B and C, Dataset EV3), with LTR retrotransposons being highly represented in this latter group (26 out of 54). To analyze TE up-regulation in D7 SAM stem cells with an independent approach, we performed qRT–PCR assays on RNA prepared from manually excised shoot apices of seedlings, as well as from sorted mCherry⁺ and mCherry⁻ nuclei of the same

material. Among 15 out of 20 TEs for which reliable data could be obtained, nine showed a significant increase, three a non-significant increase, two were unchanged, and one had less expression in mCherry⁺ nuclei compared to mCherry⁻ nuclei or non-sorted seedling samples (Fig 5A). Therefore, activation of individual TE copies in stem cells at the vegetative state was confirmed for most of candidates identified in the sequencing data. We also analyzed the sequence content of these 15 TEs (Dataset EV2) and found that, except VANDAL12 and ATHILA3, all TEs were longer than 1,900 bp

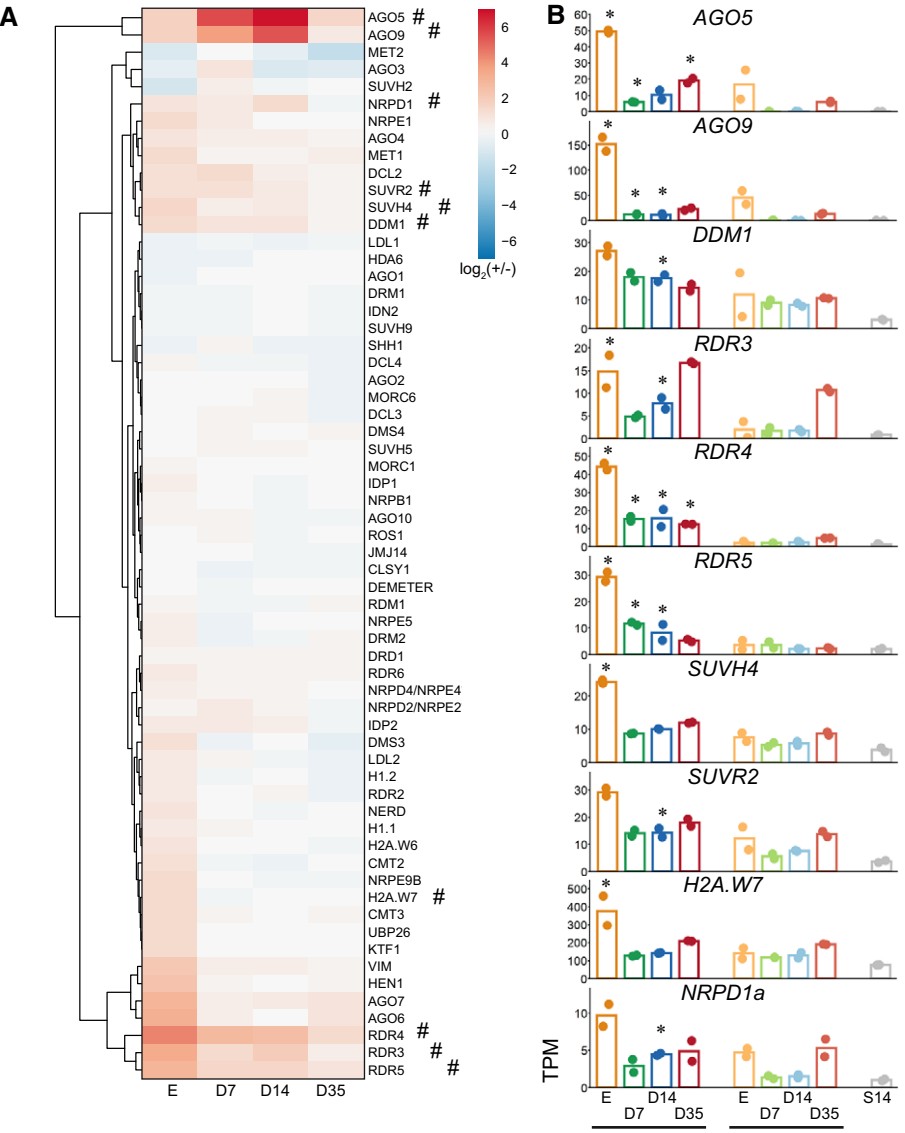

**Figure 3. Expression analysis of genes related to epigenetic regulation.**

A  Clustered expression heatmap.

B  Expression of individual significantly up-regulated genes marked with # in (A). Asterisks indicate time points of significantly different expression (Wald test, Benjamini & Hochberg corrected, $q < 0.05$) between stem and non-stem cells for each time point ($N = 2$). + = stem cells; − = non-stem cells, E = nuclei from embryos, D7/14/35 = nuclei from 7/14/35-day-old plants, S14 = nuclei from 14-day-old above-ground seedlings.

and coded for several proteins, suggesting they could be autonomous elements posing a potential threat to genome integrity.

To score the overlap of the TE with elevated expression in stem cells with those activated in different silencing mutants, we prepared RNA from shoot apices of D7 seedlings of *cmt3, dcl3, ddm1, polIV, polV, rdr2,* and *rdr6* mutants and performed qRT–PCR as for the sorted nuclei. All 12 TEs up-regulated in the stem cells were also expressed in some of the mutants, with different specificity (Fig 5B). Differential expression of six TEs in mCherry$^+$ nuclei was even more pronounced than in *ddm1* over wild type, although this mutant is well characterized for TE activation and transposition (e.g., Hirochika *et al*, 2000).

**DNA methylation changes in SAM stem cells**

To resolve the conundrum of dynamic TE expression despite increased expression of silencing-related genes, we performed DNA methylation analysis using whole-genome bisulfite sequencing of genomic DNA from D7, D14, and D35 stem and non-stem nuclei, with nuclei from 7-day- and 14-day-old whole seedlings as references. Our analysis identified methylation at CG sites (mCG) in repetitive sequences and along gene bodies, and at CHG (mCHG) and CHH sites (mCHH) (H = A, C, or T), which are mostly restricted to repetitive sequences and which are critical for TE silencing. While the mCG profile was very similar along all five chromosomes, at all

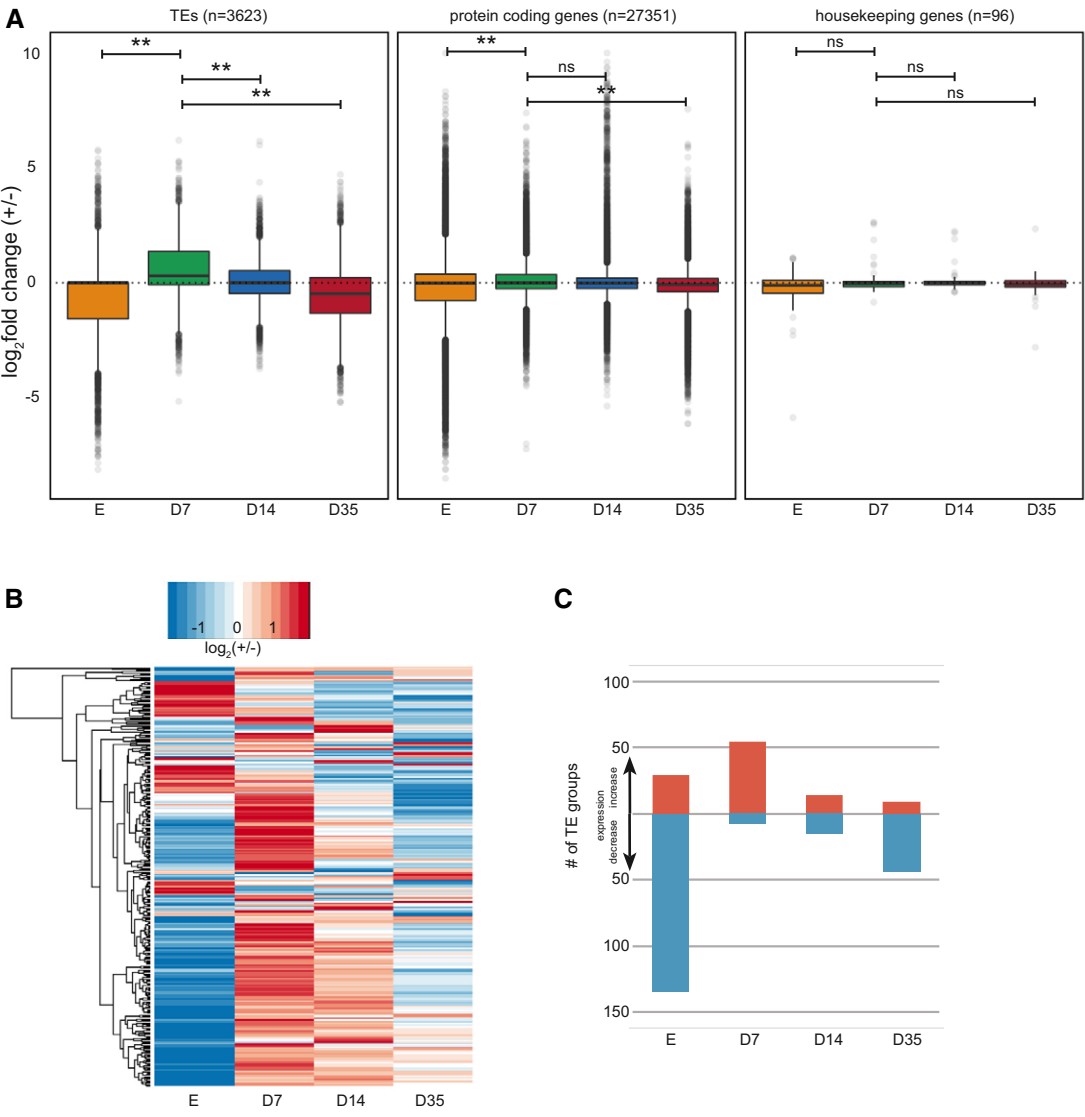

**Figure 4. Expression analysis of TEs.**

A   Box plot summarizing differential expression of unambiguously mapped individual TEs, compared with all protein-coding genes, or housekeeping genes. Box plots outline the interquartile range (IQR) with the median and whiskers ± 1.5 IQR. For all P-values, see also Table EV6. **indicates a P-value smaller than 1.01e−11 (two-sided non-paired Wilcoxon rank sum test).

B   Heatmap of expression differences for all 318 *Arabidopsis* TE families in stem cells relative to non-stem cells.

C   Number of TE families with at least 2× expression difference. E = nuclei from embryos, D7/14/35 = nuclei from 7/14/35-day-old plants.

stages and between stem and non-stem cell nuclei, there were pronounced differences around the centromeres for mCHG and mCHH, with the highest mCHG and lowest mCHH levels occurring in stem cells at D35 (Fig 6A and Appendix Fig S6). Metaplot analyses revealed that these methylation differences were found mostly at TEs, congruent with their preferential location around centromeres, while protein-coding genes were rarely affected (Fig 6B and Appendix Fig S7). mCHG levels at TEs generally increased with developmental age, reaching a maximum at D35, with TEs in stem cells having consistently higher mCHG levels than the non-stem cells at the same time point. Conversely, mCHH levels were generally lower in meristematic nuclei than in whole seedling nuclei, and

decreased with developmental age, with D35 stem cells having the lowest level of all samples (Fig 6B).

To determine whether the changes in DNA methylation from D7 to D35 correlated with TE expression, we plotted the methylation status of the 59 uniquely mapping TEs showing elevated expression in D7 stem cells (see above) over time (Fig 7A and B). In agreement with the metaplots, we observed a gradual increase of mCHG over time and a decrease of mCHH in D35 stem cell nuclei for these early-active elements. However, TEs that were not active at any time point underwent similar methylation dynamics, excluding a simple correlation between expression status at D7 and their future DNA methylation state.

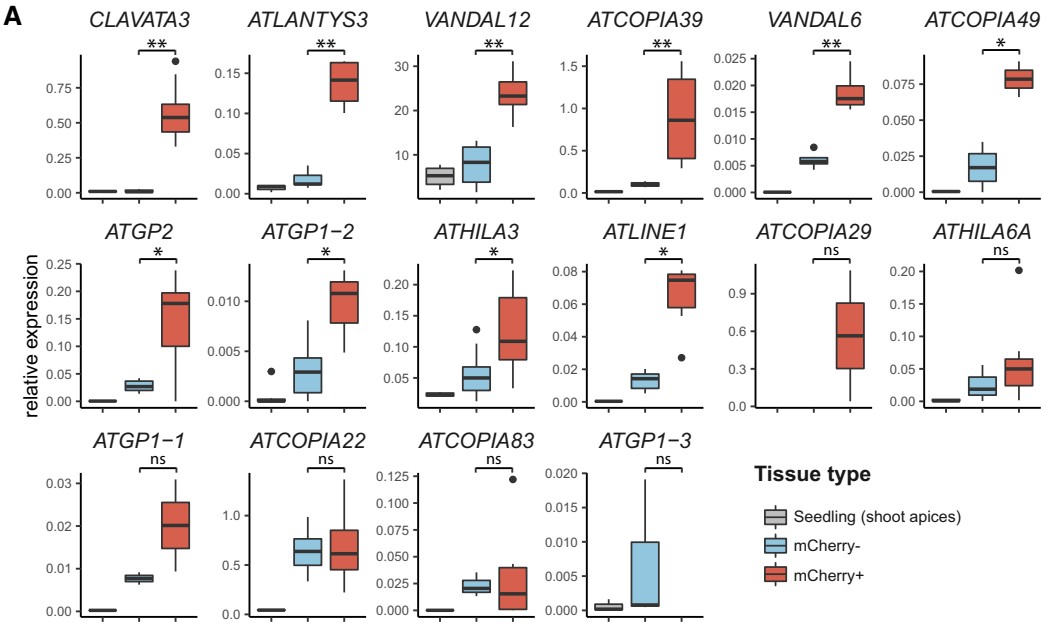

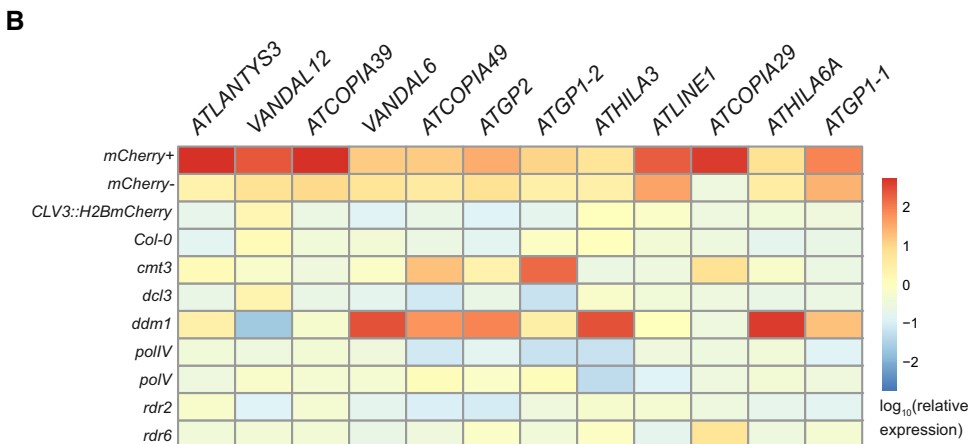

**Figure 5. Differential expression of individual TEs.**

A Increased expression of TEs in stem cells of D7 seedlings according to qRT–PCR. Box plots outline the interquartile range (IQR) with the median and whiskers ± 1.5 IQR. *N* = 2 (*ATLANTYS3*, *ATCOPIA39*, *ATCOPIA49*, *ATCOPIA29*, *ATGP1-1*), *N* = 3 (*VANDAL6*, *ATGP2*, ATGP1-*2*, *ATLINE1*, *ATGP1-3*), *N* = 4 (*ATHILA6A*, *ATCOPIA83*), and *N* = 5 (*VANDAL12*, *ATHILA3*, *ATCOPIA22*), and *N* = 6 for *CLAVATA*. *P < 0.01, **P < 0.001 (*t*-test).

B Heatmap of TE expression in several silencing mutants determined by qRT–PCR relative to a housekeeping gene. To be able to display data in log-scale also for samples with zero expression, a value of 1.0e−6 (factor 10 smaller than the smallest value) was added to all values. Data for nuclei and *CLV3::H2BmCherry* are the same as in (A). For seedling samples, *N* = 2 (*ATCOPIA49*, *ATHILA3*), *N* = 3 (*ATCOPIA39*, *VANDAL6*, *ATGP2*, *ATLINE1*, *ATCOPIA29*, *ATGP1-1*), and *N* = 4 (*ATLANTYS3*, *VANDAL12*, *ATGP1-2*, *ATHILA6A*).

## CHG methylation is mediated by DDM1 and CMT3

The chromatin-remodeling factor DDM1 was proposed to provide access for DNA methyltransferases to dense, histone H1-containing heterochromatin, and mutations in *DDM1* cause a strong reduction in DNA methylation in all sequence contexts, especially in long TEs clustered around the centromeres (Zemach *et al*, 2013). Since DDM1 expression was increased in stem cells, we asked whether DDM1 could be involved in the DNA methylation changes observed there. Indeed, plotting relative DNA methylation at TEs against TE size revealed larger mCHG differences at longer TEs (> 2.5 kb) at later stages, but with a parallel decrease in mCHH (Fig 7C). We extended the analysis to other DNA methylation factors and focused on methylation within hypomethylated DMRs (regions with less DNA methylation) that were identified in several mutants lacking different epigenetic components (Stroud *et al*, 2013). This confirmed

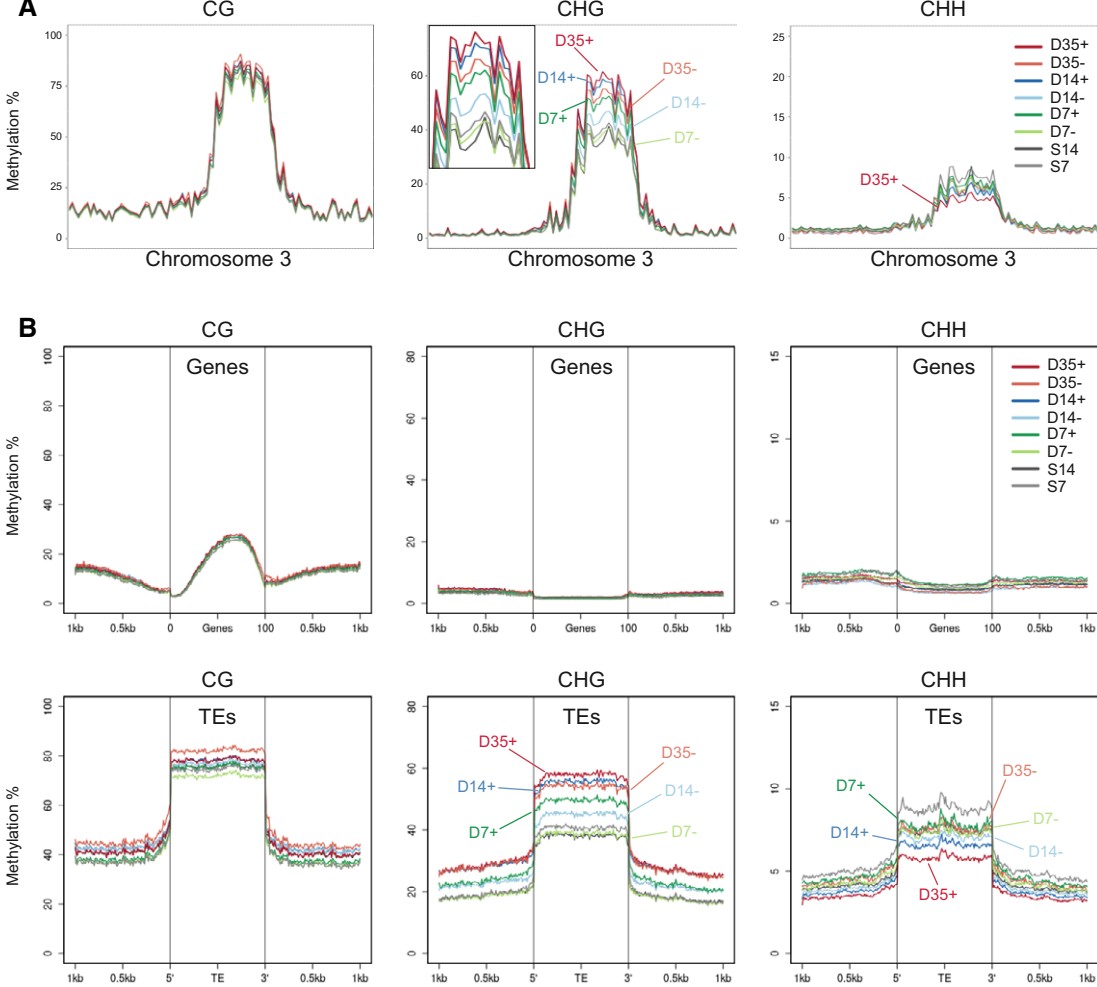

**Figure 6. DNA methylation analysis of stem cells at different developmental stages.**

A CG, CHG, and CHH methylation at chromosome 3 in stem and non-stem cells. The inset for CHG methylation shows a magnification of the pericentromeric region.

B Metaplots of DNA methylation at CG, CHG, and CHH sites for genes (top) and TEs (bottom). + = stem cells; − = non-stem cells, D7/14/35 = nuclei from 7/14/35-day-old plants, S7/14 = nuclei from 7/14-day-old above-ground seedlings.

that increased mCHG in stem cells was strongest in regions dependent on DDM1 but suggested additional involvement of the DNA methyltransferase CMT3 and the histone methyltransferases SUVH4/5/6 (Fig 8A). Regions that depend on the DNA methyltransferase MET1 and its cofactors VIM1/2/3 show less pronounced increase of mCHG methylation (Fig 8A). Therefore, DDM1, CMT3, and SUVH4/5/6 appear to contribute most to the observed mCHG increase in stem cells during development.

CMT3 and SUV4 act in a positive feedback pathway to establish and maintain mCHG (Jackson *et al*, 2002; Malagnac *et al*, 2002; Du *et al*, 2014). The stem cell-specific decrease of mCHH at D35 in SUVH4/5/6-dependent regions, contrasting the mCHG increase, was unexpected, but plausible by their overlap with CMT2 DMRs (Fig 8A). CMT2 methylates DNA in mCHH context in genomic regions that depend on DDM1 for methylation (Zemach *et al*, 2013) and may be excluded by direct competition with enhanced CMT3 activity.

## CHH but not CHG methylation converges toward meiocytes

DNA methylation control of TEs is mediated by mCHG and mCHH (Cui & Cao, 2014). The coupling of increased mCHG with decreased mCHH in D35 stem cells was inconsistent with this correlation. However, a similar discrepancy has recently been described for male meiocytes in *Arabidopsis* (Walker *et al*, 2018) in connection with reprogramming of DNA methylation in the CHH context of germline cells. Therefore, we asked whether our data for D35 meristematic cells, and especially stem cells at that stage, indicate an early convergence of CHH methylation profiles toward that found in meiocytes and whether this is involved in germline formation. We performed a principal component analysis to reveal similarities in mCHH at TEs in stem cells at each time point, including data from male meiocytes (Walker *et al*, 2018). We find that differences in mCHH between stem cells and meiocytes are smallest at D35, which is not the case for mCHG or mCG (Fig 8B and Appendix Fig S8A

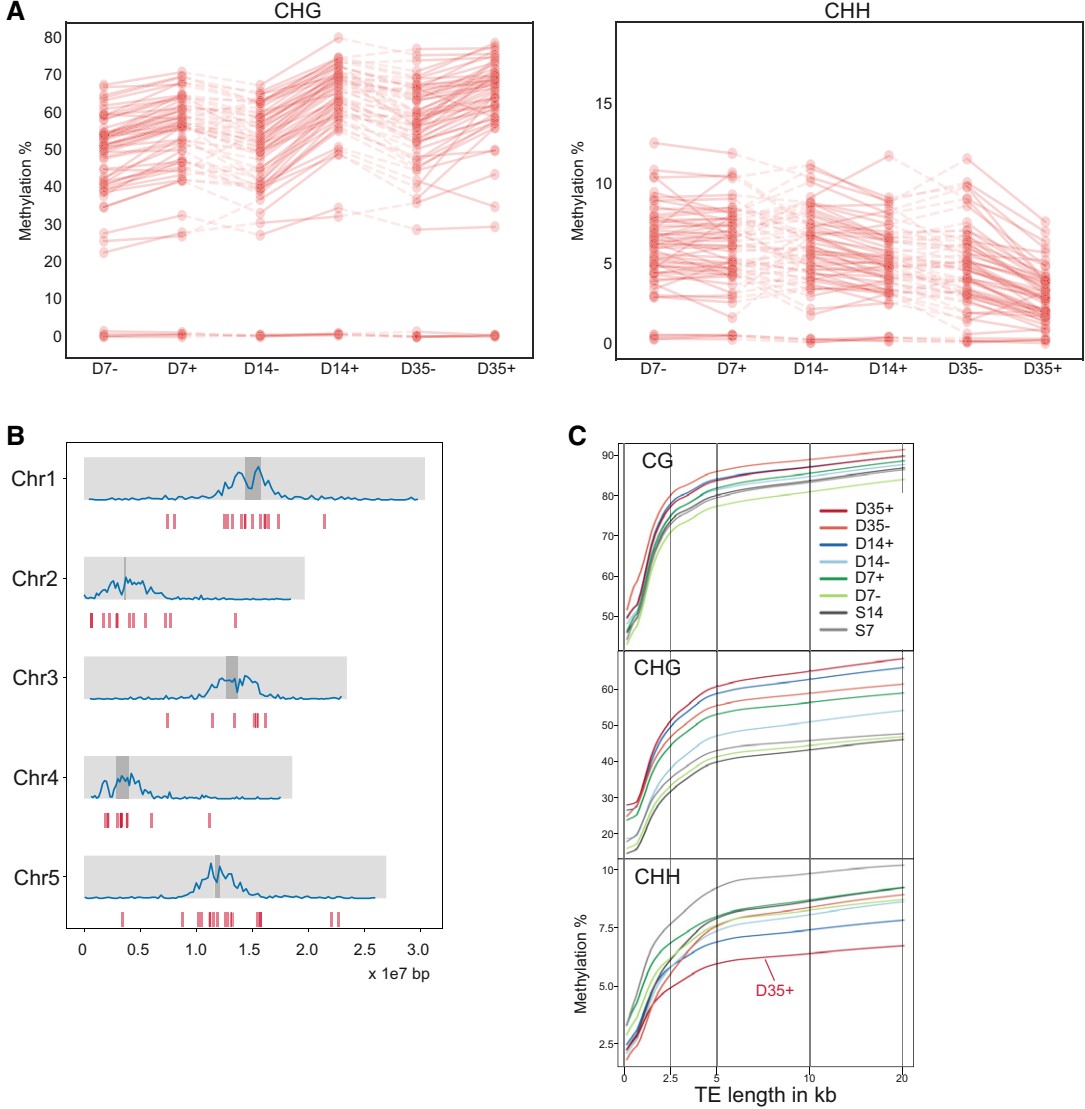

**Figure 7. DNA methylation changes at TEs.**

A   DNA methylation in % at CHG (left) and CHH (right) sites in TE with increased expression at D7, compared between developmental stages (D7/14/35 = 7/14/35-day-old plants) and nuclei of stem (+) and non-stem (−) cells.

B   Genomic location of the TEs (blue) from (A) on the five chromosomes. Dark gray bars indicate the location of centromeres, and blue lines indicate global TE density.

C   Locally weighted scatterplot smoothing fit of CG, CHG, and CHH methylation levels in stem cells and non-stem cells plotted on TE length. + = stem cells; − = non-stem cells, E = nuclei from embryos, D7/14/35 = nuclei from 7/14/35-day-old plants, S7/14 = nuclei from 7/14-day-old above-ground seedlings.

and B). Furthermore, the difference decreases from D7 and D14 along PC1 and PC2 (Fig 8B). Although the functional relevance of loss of CHH methylation in meiocytes still needs to be explored, our results point to a highly specialized control for DNA methylation in stem cells at the transition to the next generation, and several cell divisions before meiosis and gamete formation are initiated.

# Discussion

Despite the importance of SAM stem cells for plant development and reproduction, the molecular features that distinguish them from other cells are poorly understood, especially regarding their potential to transmit epigenetic information to the germline and the next generation. This is in part due to the difficulty of isolating pure cell populations of rare cell types in plants. The FANS procedure applied here overcomes this hurdle, providing sufficient numbers of cells for transcriptome and methylome analyses of pure fractions of SAM stem cells. This approach should be easy to adapt to other otherwise difficult to collect plant cell types.

We showed that the transcriptome in SAM stem cells is very similar to that in neighboring non-stem cells, with the exception of the expression of a few specific genes, and changed significantly across developmental stages, parallel to the surrounding

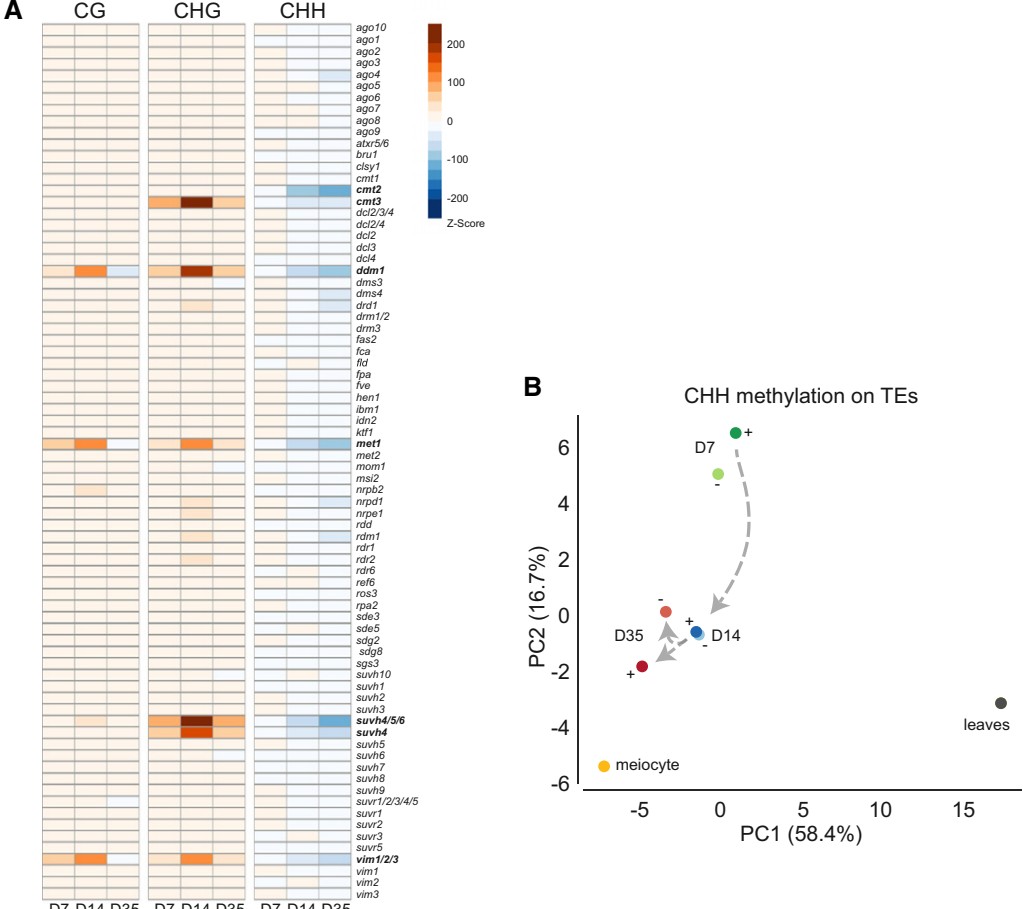

**Figure 8. Comparison with DNA methylation mutants and meiocytes.**

A  DNA methylation differences between stem and non-stem cells and their enrichment within DMRs of different epigenetic mutants using permutation tests (data from Stroud *et al*, 2013). Mutants are in alphabetical order according to the gene acronym. Red indicates an enrichment of methylated DNA in the respective context and blue a depletion.

B  Principal component analysis for relatedness between CHH DMRs at TEs in meiocytes (data from Walker *et al*, 2018) and stem (+) and non-stem (−) cell nuclei of 7-, 14-, or 35-day-old plants (D7/14/35).

meristematic cells. Therefore, SAM stem cell specification seems to be determined by positional information rather than a cell-intrinsic gene expression signature. This is plausible as plants develop largely post-embryonically. The stem cells are located at the center of a dome-shaped structure and extend through at least three cell layers that differ in cell division activity during development. They form the organ primordia for leaf or flower formation (Greb & Lohmann, 2016), following different developmental programs depending on external and internal signals.

The transcriptome data, and the gene sets showing more pronounced expression in SAM stem cells at the different time points, indicate that the stem cell developmental trajectories were initiated very early during development. For example, genes in the GO term "reproductive shoot system development" grouping are already expressed in stem cells of heart- and torpedo-staged embryos. Furthermore, the up-regulation of genes associated with cell-wall metabolism in stem cells occurs prior to the switch from a vegetative to a reproductive meristem, the time when these genes are activated in other cell types (Campbell *et al*, 2018). The

prominence of TFs among the stem cell-specific genes suggests that activation of master regulators in stem cells precedes downstream developmental decisions. Members of other functional groups, such as trans-membrane or F-box proteins, that are prominently expressed in stem cells at the various time points may be required for stem cell maintenance, signaling, or responses to external or internal signals.

Here, we have focused on the epigenetic control of TEs in stem cells, and the expression of proven or putative components of the RdDM silencing pathway. RDR4 and 5 are functionally uncharacterized, but their sequence similarity to RDR2 and 6 suggests they may have a similar, possibly stem cell-specific role. We find that AGO5 and AGO9 are specifically expressed in stem cell at three developmental stages. While they belong to different clusters of the AGO clade, both have previously been shown to be expressed in the meristematic tissue of embryos (Scutt *et al*, 2003; Havecker *et al*, 2010), and in gametes or gametophytes (Olmedo-Monfil *et al*, 2010; Tucker *et al*, 2012). AGO5 has not been linked with RdDM of TEs (Havecker *et al*, 2012), but was recently found to be associated with

24-nt siRNAs isolated from polysome fractions that match coding regions in genes and pseudogenes but rarely TEs (Marchais *et al*, 2019). DNA methylation in whole seedlings of *ago5* or *ago9* mutants is very similar to that in wild type (Stroud *et al*, 2013). However, AGO9 expression can restore DNA methylation in an *ago4* mutant if expressed from the *AGO4* promoter (Havecker *et al*, 2010), suggesting that AGO9 can substitute for AGO4 as a binding partner for heterochromatin-associated siRNAs in RdDM. We find that their expression is inversely correlated with active TEs, being lowest at D7, preceding the transition from the vegetative to reproductive stage, when diverse TE families are being transcribed, and being highest at later time points, favoring the formation of active AGO/sRNA complexes prior to flower formation. This suggests that specialized AGO members might function to provide specific protection to germline precursor cells against potentially active TEs.

The substantial and repeated epigenetic reprogramming during sexual reproduction in mammals is not recapitulated in plants, but there is accumulating evidence that de- and remethylation play a role in genome continuity between generations (reviewed in Kawashima & Berger, 2014). DNA methyltransferases are strongly expressed in plant embryos and correlate with increased *de novo* methylation (Jullien *et al*, 2012). CHH methylation at TEs in mature embryos reaches levels (> 25%) much higher than seen in other tissues analyzed (Bouyer *et al*, 2017). Furthermore, CHH methylation at TEs is highly dynamic, increasing during seed development and decreasing during germination, while CG and CHG methylation remain similar (Lin *et al*, 2017), indicating independent regulation of mCHH, mCG, and mCHG, even though the pathways overlap. Our data support this idea. We see increased CHG methylation mainly in stem cells, with lower levels in surrounding cells, and decreased CHH methylation levels. The stable levels of all methylation types across protein-coding genes points to the significance of the methylation dynamics for TE regulation.

The different tissues, developmental stages, and growth conditions used for DNA methylation analyses in other studies limit our ability to quantitatively compare data. Only one previous report in *Arabidopsis* studied DNA methylation patterns in different somatic cell types, using sorted root-tip cells (Kawakatsu *et al*, 2016). Only one cell type, the columella, a part of the root cap, showed a DNA methylome different from the other cell types. The CHH hypermethylation in these cells was explained as a result of heterochromatin decondensation, reduced expression of DDM1, and enhanced production of 24 nt siRNAs, and interpreted as strengthening silencing in the RAM stem cells located directly above the columella. Stem cells were not included in the study, but it is tempting to speculate that they might have lower CHH methylation and increased DDM1 transcription like the SAM stem cells in our study. Therefore, similar principles might protect stem cells at both end of the growth axis. However, the continuous vegetative root growth usually does not involve a developmental switch like flower induction in the shoot. The columella functions to support root-tip growth (Kumpf & Nowack, 2015), and silencing reinforcement in the meristem might be another of its roles. In the shoot, the differential expression of silencing components, the transient activation of TEs, and the observed DNA methylation dynamics may be features specific to the stem cells that promote genome protection from TEs during the period before commitment to flowering and gamete formation. Establishing high CHG and low CHH methylation at TEs in the SAM stem cells at a late developmental stage can further be seen as preparation for generating the appropriate methylation state in male meiocytes (Walker *et al*, 2018). Therefore, SAM stem cells might acquire an epigenetic status resembling germline cells long before the cytological determination of the germ cells.

Despite the high purity of our CLV3-expressing cells, it is possible that our data may be affected by heterogeneity within the SCN. This is even anticipated, as only the L2 layer of the SAM eventually gives rise to the meiocytes, the only cells that matter with regard to germline formation. Expansion of the analyses to single nuclei and additional reporters will increase the resolution and is expected to further resolve the sophisticated control over genome and epigenome stability at the transition between generations.

## Materials and Methods

### Plant material

All experiments were performed with *Arabidopsis thaliana* ecotype Col-0, wild type or transgenic for p*CLV3:H2B-mCherry*. The *pCLV3:H2B-mCherry* construct was generated as follows: The coding sequence of the *H2B* gene was PCR-amplified with primer H2B-forward and H2B-reverse (Table EV5) from cDNA prepared from 14-day-old seedlings. The vector pCLV3:erCFP (Tucker *et al*, 2008) was cut with *BamH*I and *Sac*I, and the H2B amplicon was inserted (In-Fusion, Clontech) into the open vector. The resulting plasmid was opened with *Sac*I and In-Fusion-filled with a PCR-amplified mCherry-coding fragment using the primers mCherry-fusion-F1 and mCherry-fusion-R1 (Table EV5). Correct sequence of the resulting vector pCLV3:H2B-mCherry was confirmed by Sanger sequencing. The construct was used to generate transgenic plants by the floral dip method (Clough & Bent, 1998). Primary transformants were selected with glufosinate (Merck) and their progeny screened for lines with a segregation ratio of three resistant to one sensitive plant. Homozygous offspring were propagated for seed amplification.

### Growth conditions

All plants were grown *in vitro* either on GM medium with or without selection or on soil under a 16-h light/8-h dark regime at 21°C. Material was always harvested at the same time of the light period.

### Microscopic analysis and immunostaining

For wide-field microscopy, plant material was immersed in PBS buffer and imaged with a Zeiss Axio Imager epifluorescence microscope. Isolated nuclei were imaged with an LSM780 Axio Observer, and images were deconvolved using Huygens Core (Scientific Volume Imaging) with a theoretical PSF. Immunostaining was performed according to Pasternak *et al* (2015), with an additional clearing step using ScaleA (Hama *et al*, 2011) and DAPI as counterstain. Anti-mCherry nanobodies were purchased from Chromotek (#rba594-100). Immunostains of meristems were imaged using the Airyscan mode on an LSM880 Axio Observer.

## Fluorescence-activated nuclei sorting (FANS)

For 7D/14C/35D samples, 200–800 apices (depending on size) of soil-grown plants with the corresponding age were collected. For embryo samples, ovules from siliques of a few representative plants were analyzed to contain early heart till early torpedo stage embryos, and developmentally identical siliques were used to dissect 3,000–4,000 ovules. Collected material was immediately transferred into nuclei isolation buffer on ice (NIB: 500 mM sucrose, 100 mM KCl, 10 mM Tris–HCl pH 9.5, 10 mM EDTA, 4 mM spermidine, 1 mM spermine, and 0.1% v/v 2-mercaptoethanol, prepared just before use (Pavlova *et al*, 2010)). The material was then transferred into a tube containing 1.8 ml of nuclear extraction buffer (NEB of the Sysmex CyStain® PI Absolute P kit [#05-5022] plus 1% v/v 2-mercaptoethanol) and disrupted with the TissueRuptor (Qiagen) at the lowest speed for 1 min. The suspension was filtered (30 μm filter nylon mesh, Sysmex #04-0042-2316) and centrifuged for 10 min at 4,000 *g* at 4°C. The nuclear pellet was resuspended in Precise P staining buffer (Sysmex #05-5022; plus 1% v/v 2-mercaptoethanol and DAPI to a final concentration of 5 μg/μl), incubated for 15 min, and again filtered (30 μm) into tubes (Sarstedt #55.484.001). Sorting was performed on a BD FACSAriaTM III cell sorter (70 μm nozzle). Forward/Side scatter and DAPI and mCherry gating were adjusted with wild-type nuclei (DAPI-positive, mCherry-negative) as reference. The mCherry gate was adjusted so that a maximum of 1/10 of mCherry events occurred in wild type compared to the p*CLV3:mCherry-H2B* line. For DNA extraction, nuclei were directly sorted into Genomic Lysis Buffer (Quick-DNA Microprep Kit, Zymo Research, #D3020), and DNA was purified according to the suppliers' protocol for whole blood and serum samples. DNA was quantified using pico-green on a NanoDrop fluorospectrometer (Thermo Scientific). For RNA isolation, NIB, NEB, and staining buffer were complemented with RiboLock RNase inhibitor (Thermo Scientific #EO0381, final concentration 1 U/μl) and nuclei were directly sorted into TRIzol LS (Ambion, #10296028). RNA was prepared according to the manufacturers' recommendation, except that nuclease-free glycogen (Thermo Scientific) was added during an overnight precipitation at −20°C. Amount and quality of RNA was determined on an RNA 6000 pico-chip (Bioanalyzer/Agilent Technologies). For DNA and RNA extraction, DNA-LoBind tubes (Eppendorf, #022431021) were used.

## qPCR analysis

For qPCR and enrichment analysis, RNA was extracted with TRIzol LS (Ambion) either from sorted nuclei or from shock-frozen and ground tissue material. RNA was treated with DNAse (Thermo Scientific, #79254) and reverse-transcribed with iScript (Bio-Rad, #172-5038). To quantify expression of individual TEs, 500 apices per replicate were collected from D7 plants and used for nuclei isolation and sorting of approximately 3,000 mCherry-positive and 5,000 mCherry-negative nuclei. For seedling samples (wild type and mutants), 20 apices of D7 plants were collected. Each gene was analyzed by qRT–PCR with at least two biological replicates. Extracted RNA (3 μg for seedlings and the complete preparation from collected nuclei) was digested with DNAse using the TURBO DNA-free Kit (Thermo Scientific #AM1907) in a

volume of 27 μl and 1.5 μl DNAse, then the reaction stopped with 4 μl inactivation solution. Eleven μl of DNAse-treated RNA was used to generate cDNA with Superscript IV (Thermo Scientific #18090010) and oligodT18 primers. cDNAs were diluted by addition of water, 60 μl for seedling-derived cDNA and 10 μl for nuclei generated cDNA. Two μl cDNA was used for each qPCR. Potential gDNA contamination of each cDNA was assessed using primer pairs gDNA-10 and gDNA-13 (Table EV5). qPCR assays were performed with Universal ProbeLibrary (UPL) assays (Roche, # 06402682001) with primers and probes described in Table EV5. Expression is displayed relative to the low expressed housekeeping gene AtSAND (AT2G28390).

## Library preparation and sequencing

For RNA library preparation, total RNA of biological duplicates was extracted either from nuclei directly sorted into TRIzol LS or from shock-frozen ground material and used to generate cDNA libraries with the SMART-Seq v4 Ultra Low Input RNA Kit (Clontech). For the comparison with the nuclear RNA transcriptome, RNA was extracted from DAPI-stained FANSed nuclei isolated from 14-day-old p*CLV3:mCherry-H2B* seedlings with the same protocol as for cDNA production. cDNA populations were paired-end sequenced on a HiSeq 2500 Illumina sequencing platform. For bisulfite library preparation, at least 200 pg of DNA was used. Libraries were prepared with the Pico Methyl-Seq Library Prep Kit (Zymo Research #D5456) according to the manufacturer's protocol.

## Analysis of the RNA-sequencing data

Before quantification, the appropriate adapter sequences for each mRNA-seq library were trimmed from the FASTQ files using Trim Galore (v0.5.0, https://www.bioinformatics.babraham.ac.uk/projects/trim_galore/) with automatic adapter detection. We required adapters to have a minimum overlap of four bases and only retained reads with more than 18 bases (*trim_galore –dont_gzip –length 18 –stringency 4*). For alignment of mRNA-seq data, we used the pseudoalinger Kallisto v0.45 (Bray *et al*, 2016). We first generated an index from Araport11 annotations for "non-coding gene", "novel transcribed regions", "protein-coding gene", "TE gene", and "pseudogene". We detected two transcripts that were allocated to the same AGIs but with different transcript models. Both the protein-coding transcripts AT1G06740.1 and AT3G05850.1 had different TE gene transcripts with the same unique identifier. We discarded the TE transcripts and kept the protein-coding gene annotations. Next, transcript sequences were extracted from the TAIR10 genome release using bedtools getfasta command v2.27.1 (Quinlan & Hall, 2010). The resulting fasta files as well as the H2B-mCherry sequence were concatenated and used to construct a Kallisto index. *Kallisto quant* was then run on paired-end samples using default settings.

Consecutive differential gene expression analysis was performed with DESeq2 (Love *et al*, 2014) (v1.16). Samples of the same stages were analyzed pairwise via DESeq2s Wald test (FDR < 0.05). GO enrichments were calculated using the AmiGO2 tool and the PANTHER classification system (http://amigo.geneontology.org/rte) (Mi *et al*, 2013). Visualization and clustering of the data were achieved using the R packages "gplots" and "gclus".

### DEG TE Families

All RNA-seq samples were quality-trimmed using cutadap v1.14 (Martin, 2011) and trimmomatic (Bolger *et al*, 2014) (v0.36). STAR (Dobin & Gingeras, 2015) (v2.5.2a) (Col-0 *Arabidopsis* reference genome, the Araport11 gene and TE annotations) was used as reference to map the reads, allowing multiple hits (–outFilterMultimapNmax 100 and –winAnchorMultimapNmax 100). TEtranscripts from the TEToolkit (Jin *et al*, 2015) (v1.5.1) was used in multi-mode to find DEG TE families.

### Analysis of the bisulfite-sequencing data

Illumina HiSeq 2500 sequencing data were obtained from three stages (D7, D14, and D35) each in three different settings (+: FANS-sorted stem cell tissue, −: non-stem cell but meristematic tissue, s; whole seedling). Samples D14 and D35 were sequenced with 125-bp paired-end reads and D7 with 50-bp paired-end reads (Table EV1). The data were quality-checked (fastqc) and trimmed with TrimGalore v0.4.1 (default settings with stringency = 1) and trimmomatic (Bolger *et al*, 2014) (v0.36, sliding window: 4:20, leading: 20). Bismark (Krueger & Andrews, 2011) (v0.18.1 with Bowtie2 v2.2.9) was used to map the reads to the *A. thaliana* Col-0 reference genome (including mitochondria and chloroplast genomes) in the non-directional mode with a mapping stringency of L,0,-0.6. A mapping-position-based removal of duplicates (Bismark) was applied, and the C-to-T conversion rate was calculated using the reads mapped to the chloroplast genome (ranging from 98.9 to 99.6%). Methylation was called (Bismark), ignoring the first bases according to the M-Bias plots. Samples with same stages and settings were pooled to a single sample, resulting in genome coverages for the nuclear genome from 16.4× to 53.9×. For plotting DNA methylation over the length of TEs, we used a locally weighted scatterplot smoothing fit of CG, CHG, and CHH methylation levels in stem cells and non-stem cells based on Tair10 annotation for TEs (https://www.arabidopsis.org/download/index-auto.jsp?dir = %2Fdownload_files%2FGenes%2FTAIR10_genome_release%2FTAIR10_transposable_elements).

### DMR analysis

Differentially methylation positions (DMP) were identified by Fisher's exact test. Their positions were clustered together based on a minimum distance of 50 bp between DMPs to call a differential methylated region (DMR). DMR calling was done using methylpy (https://github.com/yupenghe/methylpy.git) version 1.1.9. We used custom R and python scripts for further analysis of these DMRs. To compare the stem cell-specific methylation differences with that of methylation mutants, we calculated z-scores as follows. First, we identified DMRs between mutants and wild type. Second, we calculated differential methylation in stem cells compared to non-stem cells for cytosines overlapping with DMRs from step 1. Third, we randomly chose 1,000 genomic regions of comparable size as the DMRs in step 1, regardless whether they overlapped with TEs. Then, we calculated differential methylation between stem and non-stem cell data for these regions as in step 2. Fourth, for the distribution of these 1,000 permuted values, we calculated z-scores.

## Data availability

DNA bisulfite and RNA-seq data have been deposited in the ArrayExpress database at EMBL-EBI (www.ebi.ac.uk/arrayexpress) under accession numbers E-MTAB-5478 (http://www.ebi.ac.uk/arrayexpress/experiments/E-MTAB-5478/) and E-MTAB-5479 (http://www.ebi.ac.uk/arrayexpress/experiments/E-MTAB-5479/).

Expanded View for this article is available online.

## Acknowledgements

We thank Michael Borg, Claude Becker, Frederic Berger, and James Matthew Watson (all GMI, Vienna) for helpful comments on the manuscript, Benjamin Jaegle for help with R-scripts (GMI, Vienna), and Thomas Laux for the CLV3 promoter. We are grateful for excellent support by the GMI/IMP/IMBA Biooptics facilities, the Next Generation Sequencing and Plant Sciences units of the Vienna BioCenter Core Facilities (VBCF). We gratefully acknowledge financial support from the Austrian Science Fund (FWF I489, I1477 to O.M.S and I3687 to R.G.), Cost Action 16212 "INDEPTH" to O.M.S., and the Plant Fellows program (EU FP7) to R.G.

## Author contributions

RG and OMS conceived and designed the study and wrote the manuscript. RG, ND, AG, GB, NL, and MD performed all experiments. KR, TN, FH, and RP performed the bioinformatic analysis. MaN and MiN discussed the results and commented on the manuscript.

## Conflict of interest

The authors declare that they have no conflict of interest.

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
