## [Review Process File · The EMBO Journal]

Arabidopsis shoot stem cells display dynamic transcription and DNA methylation patterns

Ruben Gutzat, Klaus Rembart, Thomas Nussbaumer, Falko Hofmann, Rahul Pisupati, Gabriele Bradamante, Nina Daubel, Angelika Gaidora, Nicole Lettner, Mattia Donà, Magnus Nordborg, Michael Nodine, and Ortrun Mittelsten Scheid

DOI: 10.15252/embj.2019103667

Corresponding author(s): Ruben Gutzat (ruben.gutzat@gmi.oeaw.ac.at) , Ortrun Mittelsten Scheid (ortrun.mittelstenscheid@gmi.oeaw.ac.at)

Review Timeline:

Submission Date:	9th Oct 19
Editorial Decision:	15th Nov 19
Revision Received:	25th Jun 20
Editorial Decision:	7th Jul 20
Revision Received:	15th Jul 20
Accepted:	27th Jul 20

Editor: Hartmut Vodermaier

Transaction Report:

Please note that the manuscript was previously reviewed at another journal and the reports were taken into account in the decision making process at The EMBO Journal. Since the original reviews are not subject to EMBO Press' transparent review process policy, the reports and author response cannot be published.

Dr. Ruben Gutzat
Gregor Mendel Institute
Dr. Bohr-Gasse 3
Vienna, Vienna 1030
Austria

15th Nov 2019

Re: EMBOJ-2019-103667

Arabidopsis shoot stem cells display dynamic transcription and DNA methylation patterns

Thank you for submitting/transferring your manuscript together with previous reports from another journal for our editorial consideration. We have now been able to assess the study and to discuss it with a trusted arbitrating referee of our own journal, who had access to both the manuscript and to the original comments and your response to them.

As you will see from the comments below, our advisor acknowledges your isolation of SAM cell populations from different developmental stages and their combined transcriptome and DNA methylome analysis, which in our view could make the study in principle suitable for our Resource section. However, it is also apparent that some further work would be required in order to increase the amount of conclusions that may be derived from these analyses. In this respect, you will see that the arbitrator raises several persisting major concerns together with constructive proposals on how to address them, as well as a number of additional suggestions for improving more specific issues.

Should you be prepared and able to conduct additional experiments and data analyses along the lines recommended by our arbitrating referee, then we would be happy to consider this work further as an EMBO Journal Resource article. Please do not hesitate to get back to me with any questions/proposals on how to address the open points already during the early stages of revision - as it is our policy to consider only a single round of major revision. Furthermore, when revising for The EMBO Journal, please take note of our Author Guidelines for manuscript formatting, especially regarding main and "supplementary" figures and materials - in addition to 7-9 main figures, we can accommodate up to 5 "Expanded View" figures (that will also be typeset and directly available in the HTML version), with a combined "Appendix" file constituting a third level for any additional figures, tables or methods. Similarly, tables and datasets can be included in the Appendix as well as in Expanded View, depending on their relevance/format. Further information on how to prepare and upload a revised manuscript can also be found below.

We generally allow three months as standard revision time, and competing manuscripts published during this period will not negatively impact on our assessment of the conceptual advance presented by your study. Should you foresee a problem in meeting this three-month deadline, please let us know in advance and we may be able to grant an extension.

Thank you again for the opportunity to consider this study for The EMBO Journal. I look forward to hearing from you.

With best regards,

Hartmut Vodermaier, PhD
Senior Editor / The EMBO Journal
h.vodermaier@embojournal.org

- a point-by-point response to the referees' comments, with a detailed description of the changes made (as a word file).
- a word file of the manuscript text.
- individual production quality figure files (one file per figure)
- a complete author checklist, which you can download from our author guidelines (<https://www.embopress.org/page/journal/14602075/authorguide>).
- Expanded View files, replacing Supplementary Information (Please see <https://www.embopress.org/page/journal/14602075/authorguide#expandedview>)

Further information is available in our Guide For Authors:

Revision to The EMBO Journal should be submitted online within 90 days, unless an extension has been requested and approved by the editor; please click on the link below to submit the revision online before 13th Feb 2020:

Link Not Available

Referee #1:

This is an abbreviated review of the manuscript entitled "Arabidopsis shoot stem cells display dynamic transcription and DNA methylation patterns" submitted to EMBO Journal. Previous reviews were transferred with this manuscript, and are taken into consideration in this review. Overall, the strength of this work is the fact that the authors were able to get to this SAM cell population over a series of developmental landmark times, and then perform transcriptomics and DNA methylation analysis. The weakness of the manuscript is the lack of analysis beyond profiling, and how little can be firmly concluded from the profiling performed. Overall, this manuscript contributes valuable datasets to the community.

Major points from the previous reviews

1. Both previous reviewers commented on the analysis of TE expression in Figure 4. I understand that this aspect has been improved from the previous version of the paper. However, it is still difficult for the reader to understand how much increase there is in TE expression in the D7 sample compared to other control samples. Is this enough for transposition, or only a slight increase with an unknown biological role? Is this above the rate that could occur by random chance? This is an important point that deserves clarification.
 - a. The first idea is to use existing TE reporters, such as the CSHL enhancer or gene traps inserted into the exact TEs the authors find are expressed in their dataset. I understand that the authors may not want to start this experiment. It would greatly improve the work, but below is another bioinformatic approach.
 - b. qRT-PCR validation of TE expression would greatly improve the manuscript.
 - c. Detect retrotransposon intermediates via PCR
 - d. Compare TE expression in the D7 tissue with known mutants of Arabidopsis with TE expression. Is the expression as high as met1, cmt3 or ddm1 (doubtful), medium such as pol v or pol iv, barely detectable such as dcl3 or rdr2, or not present and random chance, such as rdr6?
 - e. In Figure 2 and 3 the authors show expression in SAM vs. non-SAM tissue as a series of bar graphs, which by Figure 4 the reader is accustomed to. Show these same bar plots for a series of individual TEs. This will help the reader assess the data on the individual TE-level.
2. In Figure 7A, there needs to be some sort of a negative control. If you were to take any set of TEs in the genome, they would give this same pattern. Please perform this analysis of methylation level in mutants using a control of a TE set of random elements or elements not expressed in SAM tissues.
3. In line 295 the authors argue that movement in a principal component analysis is along "a developmental trajectory". However, the principal components in this type of analysis are not defined and such a claim should not be made.

Minor points

1. Line 116 references Figure 4A. This is out of order, and I'm not even sure that the authors mean to reference this figure here.
2. Line 61-62: there are many good review articles on RdDM to reference. These are not two of them.
3. Please define the * in Figure 2A in the figure legend.
4. I realize that a previous reviewer requested the addition of the description of the up-regulated and SAM-specific genes, but this section is long (line 138-188) and very descriptive. As the reader, I found this section difficult to get through. I suggest compromising and moving much of this section

to a Supplemental Results section, and then referring to it in the main text.

5. In SAM tissue, is genic CG methylation at 100% and less (~95%) in non-meristematic tissues? Does this inform us about the location of genic DNA methylation?

6. Please zoom in and show an inset window for the CHG methylation in Figure 5A. The reader cannot resolve which line is which.

Gutzat et al.

Responses to editors and reviewers

We thank editors and reviewers for the patience with this revision. We could finish the necessary experimental work only after the Covid-19 lockdown of our institute was released.

Referee #1:

General comment

This is an abbreviated review of the manuscript entitled "Arabidopsis shoot stem cells display dynamic transcription and DNA methylation patterns" submitted to EMBO Journal. Previous reviews were transferred with this manuscript and are taken into consideration in this review. Overall, the strength of this work is the fact that the authors were able to get to this SAM cell population over a series of developmental landmark times, and then perform transcriptomics and DNA methylation analysis. The weakness of the manuscript is the lack of analysis beyond profiling, and how little can be firmly concluded from the profiling performed. Overall, this manuscript contributes valuable datasets to the community.

Thank you for appreciating the value of our work. In the revised version, we extended the molecular analysis of several transposons and confirmed their specific expression in stem cells at specific developmental stages, thereby substantiating our data and interpretation and raising interesting new perspectives.

Major points from the previous reviews

1. Both previous reviewers commented on the analysis of TE expression in Figure 4. I understand that this aspect has been improved from the previous version of the paper. However, it is still difficult for the reader to understand how much increase there is in TE expression in the D7 sample compared to other control samples. Is this enough for transposition, or only a slight increase with an unknown biological role? Is this above the rate that could occur by random chance? This is an important point that deserves clarification.

The question whether the stage-specific activation of TEs in the stem cells would allow transposition is important but difficult to address. New transposon copies integrated in the Arabidopsis genome have only been seen for heterologous elements (e.g. Lukas et al. 1995; Hirochika et al. 2000), in mutant background (e.g. Tsukahara et al. 2009; Ito et al. 2011), on the evolutionary scale (e.g. Quadrana et al. 2016), or after creating a naïve copy of an individual TE by "stripping" epigenetic marks in a mutant background (Mari-Ordonez et al. 2013). The requirements for transposition in the germline with regard to cell type, stress treatment, transcript accumulation, or other factors are not known, but we believe that technology and data from our study will allow addressing these questions in future.

a. The first idea is to use existing TE reporters, such as the CSHL enhancer or gene traps inserted into the exact TEs the authors find are expressed in their dataset. I understand that the authors may not want to start this experiment. It would greatly improve the work, but below is another bioinformatic approach.

This was a good idea, and we have tried to follow the reviewer's suggestion, but unfortunately, we could not identify a suitable gene- or enhancer trap line at stock centers for any of the TEs that were activated at D7. We were also concerned about the approach, as the gene- or enhancer trap lines are in the background of another accession (*Landsberg erecta*), which is known to differ quite substantially in its epigenetic set-up compared to Col-0.

b. qRT-PCR validation of TE expression would greatly improve the manuscript.

We have followed this suggestion and added a figure (new Fig. 5) with an extensive analysis of expression changes for several individual transposons and could validate their specific upregulation in the stem cells. Detecting expression of TEs by qRT-PCR is challenging already with ample input material, even more so with the limited number of sorted nuclei. We were successful to overcome several technical obstacles originating from these limitations. Most TE genes do not contain introns, ruling out to exclude contaminations with genomic DNA by using primers specific for cDNA. Therefore, we applied an optimized DNase digest to all samples prior to downstream analyses. We also optimized the reverse transcriptase reaction and used oligodT priming. All assays were performed with a taqman probe-based system, providing increased sensitivity and lack of background signal. We also confirmed equal expression of the reference gene in all data sets and enrichment of *CLAVATA3* expression for sorted nuclei. From 20 TEs with increased expression in stem cells at D7, five were not detectable by qRT-PCR, likely due to limiting reverse transcription. Among the other 15, nine showed a significantly and three a non-significantly higher expression in stem versus non-stem cells, their upregulation matching or even exceeding that in *ddm1*, a mutant with very high levels of TE expression. Two TEs showed no increase, and one could be detected only in non-stem cells. In summary, we validated increased TE expression in D7 stem cells in most cases. With the exception of *VANDAL12* and *ATHILA3*, all tested TEs are >1.9kb in length and code for several proteins, suggesting they are potentially functional (Supplementary Table S6).

c. Detect retrotransposon intermediates via PCR

Please see our answer to point 1: so far, we know about TE transcripts only.

d. Compare TE expression in the D7 tissue with known mutants of Arabidopsis with TE expression. Is the expression as high as *met1*, *cmt3* or *ddm1* (doubtful), medium such as *pol v* or *pol iv*, barely detectable such as *dcl3* or *rdr2*, or not present and random chance, such as *rdr6*?

As suggested, we included RNA from seedlings of the suggested mutants (*cmt3*, *dcl3*, *ddm1*, *polIV*, *polV*, *rdr2* and *rdr6*) in the expression analysis of all TEs confirmed as described in the answer to 1b. Most TEs showed strong activation also in *ddm1*, for some (*ATLANTYS3*, *VANDAL12*, *ATCOPIA39*), the upregulation was even more pronounced in stem cells (new Fig. 5b).

e. In Figure 2 and 3 the authors show expression in SAM vs. non-SAM tissue as a series of bar graphs, which by Figure 4 the reader is accustomed to. Show these same bar plots for a series of individual TEs. This will help the reader assess the data on the individual TE-level.

We followed the reviewer's suggestion and added a corresponding Supplementary Figure S8.

2. In Figure 7A, there needs to be some sort of a negative control. If you were to take any set of TEs in the genome, they would give this same pattern. Please perform this analysis of methylation level in mutants using a control of a TE set of random elements or elements not expressed in SAM tissues.

Thank you for this suggestion. We apologize that we have not been sufficiently clear about our permutation control that had been included in the previous version. We now modified the text in the Method section and the figure legend as described in the following. The figure is now Figure 8A.

We scored the enrichment of stem cell-specific methylated sites in differentially methylated regions (DMRs) for mutants. Z-scores were calculated for each DNA methylation mutant at each cytosine context as follows. First, we identified DMRs between mutants and wild-type. Second, we calculated differential methylation in stem cells compared to non-stem cells for cytosines overlapping with DMRs from step 1. Third, we randomly chose 1000 genomic regions of comparable size as the DMRs in step 1, regardless whether they overlapped with transposable elements. Then we calculated differential methylation between stem and non-stem cell data for these regions as in step 2. We consider this the better control to match our genome-wide analysis. Fourth, for the distribution of these 1000 permuted values, we calculated z-scores. We extended the figure legend accordingly:

Figure 8 | Comparison with DNA methylation mutants and meiocytes.

(a) DNA methylation differences between stem and non-stem cells and their enrichment within DMRs of different epigenetic mutants using permutation tests (data from Stroud et al. 2013) as explained in Methods. Mutants are in alphabetical order according to the gene acronym. Red indicates an enrichment of methylated DNA in the respective context and blue a depletion. (b) Principal component analysis for relatedness between CHH DMRs at TEs in meiocytes (data from Walker et al. 2018) and stem (+) and non-stem (-) cell nuclei of 7, 14, or 35 day-old plants (D7/14/35).

3. In line 295 the authors argue that movement in a principal component analysis is along "a developmental trajectory". However, the principal components in this type of analysis are not defined and such a claim should not be made.

We apologize for being imprecise and changed our wording. The new text is in line 271 ff.

Minor points

1. Line 116 references Figure 4A. This is out of order, and I'm not even sure that the authors mean to reference this figure here.

We have deleted the reference to the figure.

2. Line 61-62: there are many good review articles on RdDM to reference. These are not two of them.

We have replaced the references with two others, selected among many for currentness and accessibility.

3. Please define the * in Figure 2A in the figure legend.

An explanation was added to the legend.

4. I realize that a previous reviewer requested the addition of the description of the up-regulated and SAM-specific genes, but this section is long (line 138-188) and very descriptive. As the reader, I found this section difficult to get through. I suggest compromising and moving much of this section to a Supplemental Results section, and then referring to it in the main text.

We agree and have moved the description of genes that were more highly expressed in stem cell nuclei but had no known connection with epigenetic regulation to a separate file (Supplementary Text, with separate references).

5. In SAM tissue, is genic CG methylation at 100% and less (~95%) in non-meristematic tissues? Does this inform us about the location of genic DNA methylation?

We did not understand this question, as we never observed 100% genic methylation.

6. Please zoom in and show an inset window for the CHG methylation in Figure 5A. The reader cannot resolve which line is which.

Thank you for the suggestion, we have done this in (now) Fig. 6a.

Dr. Ruben Gutzat
Gregor Mendel Institute
Dr. Bohr-Gasse 3
Vienna, Vienna 1030
Austria

7th Jul 2020

Re: EMBOJ-2019-103667R

Arabidopsis shoot stem cells display dynamic transcription and DNA methylation patterns

Thank you for submitting your revision to The EMBO Journal. I have now had a chance to carefully assess your responses to our arbitrating referee as well as the revised manuscript, and am pleased to inform you that we would like to proceed further with publication of this work, following final modifications to incorporate various editorial points as detailed below.

- Pre-acceptance checks by our data editors have raised several queries with the data descriptors in the figure legends, which you will find as comments in the attached edited/commented Word documents with activated "Track changes" option. I would appreciate if you incorporated the requested final text modifications and answered the Figure legend queries directly in this version (and modified figures where necessary), uploading the edited main text document upon resubmission with changes/additions still highlighted via the "Track changes" option, to facilitate our final checking.
- Please remove figure legends from the main figure files - the legends should only be in the main text.
- Please refer to our author guide (www.embopress.org/page/journal/14602075/authorguide#expandedview) regarding supplemental material, and consider re-organizing the figures and tables currently included as "supplementary" in the Appendix:
 - Appendix figures should be renamed to "Appendix Figure S1/2/3..." and referenced in the text as such; their legends should be removed from the main text and only appear in the Appendix
 - Up to 5 Appendix Figures may be promoted to Expanded View Figures (callout: "Figure EV1/2/3") - in this case, they should be uploaded as individual figure files like the main figures, and their legends amended to the main text (Expanded View Figures will be type-set and directly visible/expandable in the HTML version of the article).
 - Except for Appendix Table S1, I think all included tables/datasets would be much better suited as Expanded View Tables (Tables S3-5, S8-10) or Expanded View Datasets (Tables S2, S6, S7). In this case, they could be available as multi-tab Excel spreadsheets. Their respective legends should in each case be included in an additional, separate "Legend" tab; and their in-text call-outs changed to "Table EV1/2/3" or "Dataset EV1/2/3", respectively.
- Please make sure to indicate the re-use/re-publication of the FACS gating examples in Appendix Figure S1 in the respective figure legend, and state a valid reprint permission at this point.

I am therefore returning the manuscript to you for a final round of minor revision, to allow you to make these adjustments and upload all modified files. Once we will have received them, we should be ready to proceed with formal acceptance and production of the manuscript. Please do not hesitate to contact me should have any questions regarding the final modifications/format requirements.

With best regards,

Hartmut Vodermaier, PhD
Senior Editor / The EMBO Journal
h.vodermaier@embojournal.org

- a point-by-point response to the referees' comments, with a detailed description of the changes made (as a word file).
- a word file of the manuscript text.
- individual production quality figure files (one file per figure)
- a complete author checklist, which you can download from our author guidelines (<https://www.embopress.org/page/journal/14602075/authorguide>).
- Expanded View files, replacing Supplementary Information (Please see <https://www.embopress.org/page/journal/14602075/authorguide#expandedview>)

Further information is available in our Guide For Authors:

Revision to The EMBO Journal should be submitted online within 90 days, unless an extension has been requested and approved by the editor; please click on the link below to submit the revision online before 5th Oct 2020:

Thanks for the good news! We are very happy and grateful for your, your colleagues' and the reviewer's help to get this work into an optimal shape.

The new version of the Manuscript plus texts (using the file you sent us) with track changes was uploaded via the EMBO submission system. All Figure legends were removed and we followed all your suggestions regarding Tables and Datasets. We also assigned 3 Figures from the Appendix as Expanded View Figures (EV1-3). We state the reprint permission of (now) Figure EV1 and in the attachment you will find our license agreement with Springer.

Please do not hesitate to contact Ortrun or me in case something is not clear.

Looking forward to hear from you!

With best wishes,

Ruben Gutzat and Ortrun Mittelsten Scheid

Dr. Ruben Gutzat
Gregor Mendel Institute
Dr. Bohr-Gasse 3
Vienna, Vienna 1030
Austria

27th Jul 2020

Re: EMBOJ-2019-103667R1

Arabidopsis shoot stem cells display dynamic transcription and DNA methylation patterns

Thank you for submitting your final revised manuscript files for our consideration. I am pleased to inform you that we have now accepted it for publication in The EMBO Journal.

Your article will be processed for publication in The EMBO Journal by EMBO Press and Wiley, who will contact you with further information regarding production/publication procedures and license requirements. You will also be provided with page proofs after copy-editing and typesetting of main manuscript and expanded view figure files.

Should you be planning a Press Release on your article, please get in contact with embojournal@wiley.com as early as possible, in order to coordinate publication and release dates.

With kind regards,

Hartmut Vodermaier, PhD
Senior Editor / The EMBO Journal
h.vodermaier@embojournal.org

Corresponding Author Name: Ortrun Mittelsten Scheid, Ruben Gutzat

Manuscript Number: EMBOJ-2019-103667